# CAN DIFFERENTIABLE DECISION TREES LEARN INTERPRETABLE REWARD FUNCTIONS?

## ABSTRACT

There is an increasing interest in learning reward functions that model human preferences. However, many frameworks use blackbox learning methods that, while expressive, are difficult to interpret. We propose and evaluate a novel approach for learning expressive and interpretable reward functions from preferences using Differentiable Decision Trees (DDTs). Our experiments across several domains, including CartPole, Visual Gridworld environments and Atari games, provide evidence that that the tree structure of our learned reward function is useful in determining the extent to which the reward function is aligned with human preferences. We provide experimentall evidence that reward DDTs can achieve competitive performance when compared with larger capacity deep neural network reward functions. We also observe that the choice between soft and hard (argmax) output of reward DDT reveals a tension between wanting highly shaped rewards to ensure good RL performance, while also wanting simpler, more interpretable rewards.

## 1 INTRODUCTION

The reward function is central to reinforcement learning (RL) algorithms (57); however, it is often difficult to manually specify a precise reward function for diverse tasks (46; 37), motivating learning reward functions from human input (1; 51; 15; 12; 8; 5; 47). In this paper, we focus on the problem of learning interpretable reward functions.

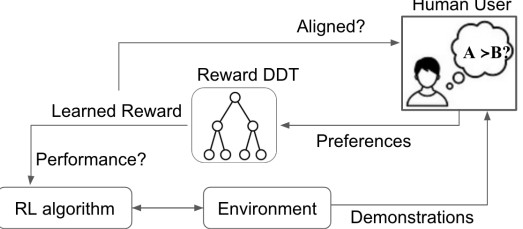

Figure 1: We propose an approach for training interpretable reward functions using differentiable decision trees. Human feedback in the form of trajectory preferences is used to efficiently train the reward model end-to-end. The tree structure allows the human to evaluate the alignment of the learned reward function.

Most modern reward learning methods use deep neural networks (19; 15; 30; 32; 60). However, despite the growing interest in explaining these black box methods (26; 63; 29; 49), deep neural networks remain extremely difficult to interpret. In the context of reward learning, it is especially critical that we can interpret the learned objective—if we can not understand the objective that a robot or AI system has learned, then it is difficult to know if the AI's behavior will be aligned with human preferences and intent (50; 40; 14). This is particularly significant in tasks where human safety is on the line, for example in healthcare, autonomous navigation, and assistive robots.

Thus, we are faced with a problem: we want highly accurate and expressive reward models, but we also want to be able to interpret the learned reward function. A natural step towards both of these goals is to combine the expressiveness of neural networks with an architecture choice that is easy for a human being to interpret, such as a decision tree. To tackle the the aforementioned problems, we propose a novel reward learning approach that uses an end-to-end differentiable decision tree model for learning interpretable reward functions from pairwise preferences. We evaluate our approach on three different domains: CartPole (11), a novel set of Visual MNIST Gridworld environments, and two Atari games from the Arcade Learning Environment (3). Our results demonstrate the ability to learn expressive and interpretable reward functions from both low- and high-dimensional state inputs.

Learning a reward model as a differentiable decision tree (DDT) has the advantage that the tree structure explicitly breaks the reward prediction for a state into a finite number of routing decisions within the tree. This provides the potential to understand how the reward predictions are being made. Our framework generates global explanations for all inputs across both low- and medium-dimensional environments such as CartPole and visual MNIST gridworlds. For high-dimensional visual state space, such as Atari, we propose a novel form of hybrid explanation that approximates global explanations by leveraging aggregations of individual input states. Our results provide evidence that we can leverage the interpretability of the learned reward DDT to identify reward misalignment.

This paper makes the following contributions: (1) We introduce a reward learning framework (Fig 1) that employs differentiable decision trees to learn human intent using trajectory preference labels without necessitating any hand-crafting of the input feature space. (2) We propose hybrid explanations for internal nodes that approximate of global explanations by leveraging aggregations of individual input states. (3) We study the ability of Differential Decision Trees (DDTs) to learn interpretable rewards across several domains including complex visual-control tasks and find that Reward DDTs can learn an interpretable reward function with RL performance comparable to that of a black-box neural network. *To the best of our knowledge, our framework is the first interpretable tree-based method for reward learning that can be applied in visual domains.*

## 2 RELATED WORK

**Preference Learning**    Preference learning (61) is applicable across multiple forms of human input: prior work has shown that demonstrations (13), e-stops (24), rankings (47), and corrections (43), can all be represented in terms of pairwise preferences. Thus, our approach is applicable, even when pairwise preference labels are not explicitly available. Prior work on learning reward functions from preference labels typically either assumes access to a set of hand-designed reward features (52; 6; 43; 24) or uses deep convolutional or fully connected networks for reward learning (15; 12; 38; 32; 55; 41). By contrast, we study the extent to which we can learn expressive, but also interpretable reward functions from preferences via differentiable decision trees (22).

**Explaining and Interpreting Reward Functions**    In the past few years, various attempts have been made to understand learned reward functions. Prior work compares learned reward functions to a ground truth reward using a pseudometric (27), uses saliency maps and counterfactuals (12; 44; 42), leverages human teaching strategies (39; 9), wrapper modules (34) or uses human-centric evaluation methods for reward explanation (53). Some of the prior works have also looked at using expert-driven reward design techniques to incorporate structural and interpretability constraints (33; 17; 31).Recent work shows that reward functions learned from human preferences via deep neural networks often suffer from spurious correlations and reward misidentificiation (60). In this work we seek to investigate to what extent differentiable decision trees enable interpretable reward functions that can aid in detecting reward misidentification.

**Differentiable Decision Trees**    Differential Decision Trees (DDTs), also referred to as Soft Decision Trees, seek to combine flexibility of neural networks with interpretable structure of decision trees. DDTs have been previously applied to supervised learning tasks (22; 59; 28) and unsupervised tasks (62). Recent work has investigated using DDTs for reinforcement learning tasks (56; 16; 58; 18; 48), but focus on *policy learning* using DDTs. Compared to prior work, the primary objective of our work is to *learn interpretable reward functions* using DDTs. While policy explanations are important, they only show what triggers an agent to take an action, rather than the reason for taking the action. By understanding agent's reward, we gain insight into the agent's value alignment (40; 20; 14) which can transfer across different embodiments and dynamics (23), unlike policies. Furthermore, prior work using DDTs for policy learning only considers low-dimensional, non-visual inputs (56; 16). By contrast, we study DDTs applied to high-dimensional image observations.

**Decision Trees for Reward Learning**    There has been very little prior work on using decision trees for reward learning. Bewley et al. (5) formulate a tree-based reward learning method that requires a complex, non-differentiable, multi-stage optimization procedure. By contrast, our approach is end-to-end differentiable and trainable using a simple cross entropy loss. Bewley et al. (5) only consider low-dimensional inputs where internal nodes in tree have the form $(s, a)_d \geq c$ for each dimension $d$

of the state-action space and threshold $c$. This approach divides state-action space into axis aligned hyperrectangles, which works for lower-dimensional spaces, but does not scale to higher-dimensional state and action spaces. More recent work uses a differentiable loss function but is not end-to-end differentiable as it requires reward tree to regrow at each update (4). Furthermore, prior work requires hand crafting input features per decision node in the decision tree which makes it intractable to scale to the types of visual inputs we consider. Our work seeks to address these limitations by learning reward function DDTs that are end-to-end differentiable, do not require hand-crafted features, and scale easily to high dimensional pixel inputs.

# 3 Reward Learning using Differentiable Decision Trees

Classical decision trees are interpretable and easy to tune (36; 45); however, they require feature engineering which can result in lower performance and less generalization compared with other machine learning approaches (22; 28). In this section, we discuss our proposed approach for learning interpretable but expressive reward functions via differentiable decision trees (DDTs).

## 3.1 Internal Nodes

Classical decision trees consist of internal nodes that deterministically route inputs. Since we want our reward function tree to be easily trained using backpropagation, we define a differentiable soft routing function that retains the expressiveness of a neural network by learning the routing function for each non-leaf node. We define an internal node in the DDT as a sequence of one or more parameterized functions applied to the input to the DDT to determine probability of routing left or right. Because each internal node depends directly on the input, the differentiable decision tree learns a hierarchy of decision boundaries that determine the routing probabilities for each input. We describe two variants of an internal node below:

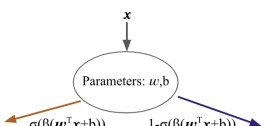

Figure 2: Routing probability of an internal node in a DDT.

**Simple Internal Node**   Proposed by Frosst and Hinton (22), a simple internal routing node, $i$, has a linear layer with learnable parameters $\mathbf{w}_i$ and a bias term $b$ upon which sigmoid activation function, $\sigma$, is applied to derive the routing probability given an input $\mathbf{x}$ (Fig 2). Thus, the probability at node $i$ of routing to the left branch is defined as $p_i(\mathbf{x}) = \sigma(\beta(\mathbf{x} \cdot \mathbf{w}_i + b))$. An inverse temperature, $\beta$, is included in the equation above for controlling the degree of soft decisions.

**Sophisticated Internal Node**   For higher-dimensional inputs we propose an alternative internal node architecture, which consists of a single convolutional layer with Leaky ReLU as the non-linearity followed by a fully connected linear layer, as before. The probability of going to the leftmost branch at an internal node $i$ is defined as $p_i(\mathbf{x}) = \sigma((\text{LeakyReLU}(\text{Conv2d}(\mathbf{x}))) \cdot \mathbf{w}_i + b)$.

## 3.2 Leaf Nodes

Following prior work that uses DDTs for classification problems (22), a leaf node $l$ is parameterized by $\phi^l$, that defines a softmax distribution over a discrete number of reward outputs $c$. The probability distribution at a leaf, $\boldsymbol{Q}^l$, is defined as $\boldsymbol{Q}_i^l = \exp(\phi_i^l)/(\sum_{j=0}^c \exp(\phi_j^l))$. We propose and study two ways to obtain rewards at the leaves of a reward DDT.

**Multi-Class Reward Leaf (CRL)**   This kind of leaf node performs multi-class classification and assumes that the user can specify classes of possible discrete reward values suitable for the given task, such that reward vector is of the form $\mathbf{R} = (r_1, r_2, \ldots, r_c)$, where $c$ denotes the number of classes for the DDT, where each class index $i$ is assigned reward value $r_i$. The learnable parameters $\phi^l$ at leaf $l$ form the logit values of a classification problem over the possible reward values in $\mathbf{R}$.

**Min-Max Reward Interpolation Leaf (IL)**   As an alternative to the classification approach, we model the reward of a DDT as *regression problem*, that requires the user to specify only the minimum and maximum range of possible reward values as opposed to requiring finite set of possible reward values as in CRL. Thus, the reward vector is of the form $\mathbf{R} = (R_{\min}, R_{\max})$, where $R_{\min}$ and $R_{\max}$

correspond to minimum and maximum of the user-defined range of rewards respectively. Given this parameterization, we can view the reward output of a DDT as a convex combination of $R_{\min}$ and $R_{\max}$ based on the learned parameters $\phi^l$.

## 3.3 TRAINING DDTs FOR REWARD LEARNING USING HUMAN PREFERENCES

As we want our reward DDT to be end-to-end differentiable when learning a reward function from preference labels,we formulate soft reward prediction as follows. First, the tree of depth $d \geq 1$ is built by $\sum_{k=0}^{d-1} 2^k$ internal nodes and $2^d$ leaves. Given an input $\mathbf{x}$, we denote the path probability from root node to a leaf $\ell$ by $P^\ell(\mathbf{x})$.The soft reward prediction of the tree is given by the sum over all leaves of the path probability of reaching that leaf $P^\ell(\mathbf{x})$ , multiplied with the soft reward at the leaf:

$$r_\theta(\mathbf{x}) = \sum_\ell P^\ell(\mathbf{x})(\mathbf{Q}^\ell \cdot \mathbf{R}) . \tag{1}$$

To train our reward function DDT, we propose to leverage pairwise preference labels over trajectories. Given preferences over trajectories of the form $\tau_i \prec \tau_j$, where $\tau = (s_1, s_2, ...s_T)$, we can train our entire differentiable decision tree via the following cross entropy loss resulting from the Bradley Terry model of preferences (10; 15). For DDTs with sophisticated nodes, we also regularized the network to ensure that, on average across many inputs, each internal node routes left and right equally often (see Appendix A for details).

## 3.4 USING A TRAINED REWARD DDT FOR REWARD PREDICTION

To use the trained reward DDT for reward prediction at test time (e.g., when using the reward DDT for reinforcement learning), one option is to use the soft reward (averaged across all leaf nodes weighted by routing probability); however, this loses interpretability since we cannot trace the predicted reward to a small number of discrete decisions. To enable interpretable reward predictions we can alternatively output a single reward prediction by first finding the leaf node with maximum routing probability for a given input $\mathbf{x}$:

$$l^* = \arg \max_{l \in L} P^\ell(\mathbf{x}) , \tag{2}$$

where $L$ denotes set of all leaf nodes in the DDT. And then the test-time output of a reward DDT with CRL Leaf nodes is given as $r_{max}(\mathbf{x}) = r_i$, for $i = \arg \max_i \mathbf{Q}_i^{\ell^*}$; while for IL Leaf nodes it is given as $r_{max}(\mathbf{x}) = \mathbf{Q}^{\ell^*} \cdot (R_{\min}, R_{\max})$.

## 4 EXPERIMENTS AND RESULTS

We evaluate reward DDTs on three different types of environments: CartPole, a novel set of MNIST Gridworld environments, and Atari 2600 games (2). In Appendix D.3, we examine the effect of choice of internal node architecture on the interpretability of the reward DDT. We find that in low and medium-dimensional state space, both type of nodes perform equivalently but as we scale up to higher dimensions such as Atari, sophisticated internal nodes perform better.

## 4.1 CARTPOLE

The Cartpole environment comprises a cart with a pole attached to it, sliding on a friction-less track. The objective is to balance the pole on the cart for as long as possible while cart moves to left and right along the track without letting pole fall beyond $\pm 12°$ from the upright position and without letting the cart move beyond $\pm 2.4$ units along the track. We assume no access to ground truth reward and we use our DDT framework to learn ground-truth reward from pairwise preference labels.

**Setup** To train a reward function DDT, we generate a wide variety of trajectories by running a random policy in the environment for 200 steps for each trajectory with no access to any kind of terminal or done flag (since this would leak significant information about the true reward (21)). Thus, we ignore the done flag in the Cartpole environment and keep accumulating states in the trajectory for 200 timesteps, even if the pole falls over. We design a synthetic preference labeler that returns pairwise preferences based on the true (but unobserved) reward.

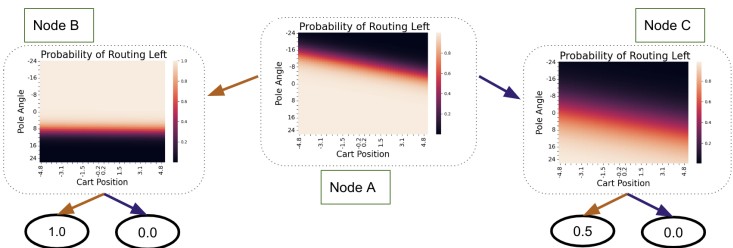

Figure 3: **CartPole Reward DDT.** The heatmap for each internal node depicts the learned routing probability.Leaf nodes are depicted as circular nodes with their soft reward values. The tree learns that small magnitude pole angles are good and should be routed to a +1 reward but there is no learned decision boundary that clearly captures the preference that cart position stay within the range $[-2.4, 2.4]$ showing that learned reward is mis-aligned.

|  | DDT | | Baseline |
|  | CRL Soft | CRL Argmax | Neural Network |
|---|---|---|---|
| Mean | 189.90 | 183.466 | 66.29 |
| Standard Deviation | 10.32 | 17.39 | 66.70 |

Table 1: **Evaluating RL on Learned Reward Function in CartPole.** DDTs with soft outputs outperforms argmax rewards at test time and both significantly outperform RL performance of a non-interpretable fully connected 2-layer reward network baseline. CRL denotes a tree with Class Reward Leaf nodes. The table shows Mean and Standard deviation across 10 seeds averaged over 100 rollouts.

Given pairwise preference labels over these suboptimal trajectories, we train a reward DDT of depth 2: the tree has 3 internal nodes and 4 leaf nodes (we experimented with larger depth and did not see any improvements in performance). We use multi-class reward leaf(CRL) nodes with 2 classes: $\mathbf{R} = (0.0, 1.0)$. This is because the true reward is binary and we wish to compare the results of the reward DDT to the ground truth reward.

It is important to note that even though the ground truth preferences are based on both cart position and pole angle, the pole usually falls past the desirable range long before the cart leaves the desirable range. Thus, our dataset is biased and we hope to be able to pickup on this bias, and the corresponding misaligned reward function by inspecting the learned reward DDT. To evaluate RL performance of the learned reward function by the DDT, we train a vanilla policy gradient agent on the learned reward function to obtain the final policy and then evaluate this learned policy on the ground-truth reward function. We also compute the ground-truth performance of a policy gradient policy trained on the same dataset using a neural network reward function comprised of 2 fully connected layers.

**Results** The results in Table 1 show that a simple reward DDT outperforms a neural network made up of fully-connected layers, irrespective of whether the policy is learned using soft rewards or using the maximum probability path across the learned reward DDT. Fig 3 shows learned reward DDT. Because the input space to the reward function is 2-dimensional (cart position and pole angle) we visualize the heatmap of routing probability at each internal node (as a function of cart position and pole angle) along with leaf distributions. From DDT it is clear that most of the routing decisions are made based on pole angle, rather than cart position. A nice feature of the reward DDT is that we can easily visually interpret the learned reward just by looking at the tree. From Fig 3 we see that while the tree learns that small magnitude pole angles are good and should be routed to a +1 reward, there is no learned decision boundary that clearly captures the preference that cart position stay within the range $[-2.4, 2.4]$.

## 4.2 MNIST Gridworlds

Next, we use our reward DDT framework for solving MDPs with visual inputs. We propose three novel gridworld environments where agents can move in the four cardinal directions and each state is

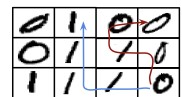

(a) Pairwise trajectory preference

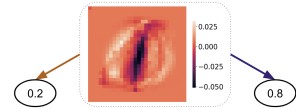

(b) Visualization of Learned Reward DDT

Figure 4: **MNIST (0/1) Gridworld.** (a) A pair of trajectories with the same starting state, where the blue trajectory (which visits more 1's) is preferred over the red trajectory. (b) Heatmap of Learned Reward DDT : The dark pixels at center of heatmap form an approximate shape of digit 1 and are routed to right as the dark colors in heatmap mean that those pixels are turned off, while lighter pixels represent shape of digit 0 and routed to left as those pixels are turned on. Leaf nodes are depicted as circular nodes with their soft reward values.

| | DDT | | | | Baselines | |
| | CRL Soft | CRL Argmax | IL Soft | IL Argmax | NNet | Random |
|---|---|---|---|---|---|---|
| MNIST 0-1 | 92.37% | 82.27% | 99.98 | 100% | 98.2% | 7.38% |
| MNIST 0-3 | 71.66% | 71.66% | 98.99% | 97.77% | 99.53% | 7.56% |
| MNIST 0-9 | 62.15% | 62.15% | 97.32% | 92.87% | 97.74% | 7.93% |

Table 2: RL Performance as the percentage of expected return obtained relative to the performance of an optimal policy on the ground-truth reward. Results are averaged across 100 different MDPs. We find that reward DDTs with Interpolated Leaf nodes (IL) perform as well as a neural network reward in both gridworld environments, while using Class Reward Leaf nodes (CRL) results in lower performance, but still significantly outperforms a random policy (Random). This provides evidence that our simple framework can learn interpretable reward without losing the expressiveness of a neural network under RL evaluation.

associated with a $28 \times 28$ grey-scale image of the MNIST digit and the value of the digit determines the true unobserved reward at that state. The rewards must be inferred from preferences over pairwise demonstrations, where preference label assignment is based on comparison between sum of ground truth labels of each state in a demonstration. To interpret the learned reward DDT, we construct a pixel-level activation heatmap for each internal node by starting with a blank image and iteratively toggling on and off each pixel and computing the resulting difference in routing probabilities for each internal node. We compare the performance of the RL policy optimized using the learned DDT reward function against the optimal policy under the ground truth reward, a random policy, and a policy learnt by a neural network trained on the same preference dataset.

### 4.2.1 MNIST (0/1) GRIDWORLD

**Setup** In this 5x5 gridworld each state in the MDP corresponds to a MNIST digit 0 or 1. To test whether we can learn an interpretable reward function from pairs of preference demonstrations over trajectories (see Fig 4a for an example pairwise trajectory comparison), we modeled the reward as a DDT of depth 1 with one simple internal node as the root node and 2 CRL leaf nodes with reward vector $\mathbf{R} = (0.0, 1.0)$. See Appendix C for more details.

**Results** The resulting heatmap in Fig 4b demonstrates the fact that DDT learns to branch based on visually interpretable features that correspond to a 0 (routes to left leaf node) and 1 (routes to right leaf node).The RL performance using the Soft Reward from CRL Leaf DDT on MNIST 0-1 environment is shown in Table 2 is comparable to a deep neural network reward function trained on pairwise preferences. We observe that taking the maximum probability path across the learned reward tree results in a small loss of performance relative to when we take soft reward from the learned DDT.

### 4.2.2 MNIST (0-3) GRIDWORLD

**Setup** We increased the complexity of the reward function in the 5x5 gridworld MDP such that the states now correspond to MNIST digits 0,1,2,3. We trained two different types of reward DDTs of depth 2 with 3 simple internal nodes and 4 leaf nodes of type : CRL with $\mathbf{R} = (0, 1, 2, 3)$ and IL with $R_{\min} = 0$ and $R_{\max} = 3$ (see Appendix D.

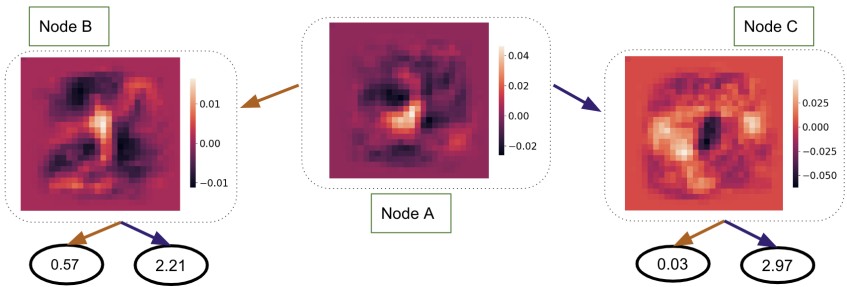

Figure 5: **Visualization of MNIST (0-3) Reward DDT** The activation maps provide interpretability and show that images of 1s are routed left to node B and then left to leaf node that outputs 0.53. Images of 0s are routed from Node A to node C then to the 0.05 leaf node. Images of 2s are routed from A to B then to 2.14. Images of 3s are routed from A to C then to 2.96.Leaf nodes are depicted as circular nodes with their soft reward values.

**Results** In Appendix D we visualize and compare the reward DDTs with CRL and IL leaf nodes and find that in CRL formulation the leaf nodes fail to specialize and argmax output of the leaf nodes is either 0 or 3, despite investigating several regularization techniques (see Appendix D). We conclude that using IL leaf nodes is best when learning complicated reward functions where we wish to output more than two possible rewards. It is also simpler,as it requires the human to only specify a range of desired reward values, $[R_{\min}, R_{\max}]$. Thus, we focus on our analysis of the interpretability of the IL reward DDT.

In Fig 5, the activation heatmaps are not simply combinations of digits like Fig 4, but rather isolate pixel features that are maximally discriminative . However, they still provide a strong understanding of what the network has learned and how it predicts rewards. These heatmaps show that DDT learns to route based on visual representations of each digit: Node B routes left for vertical pixels in center from vertical stroke of digit 1 and sends 1's left while using upper and lower curves of digit 2 to route 2's right (note the black shadow that looks like a 2). To discriminate between a digit 0 and 3, node C discriminates based on middle cusp of 3 and left curve of the 0. Finally, node A learns that what best distinguishes 1s and 2s from 0s and 3s is presence of central lower pixels—the highest activation for node A is intersection of the 1 and 2 which falls between middle and lower cusps of 3 and inside digit 0. Note that tree uses min-max reward interpolation between $R_{min} = 0$ and $R_{max} = 3$. Note that there are no explicit reward labels, just pairwise preferences over trajectories. Despite the lack of fine-grained feedback, the DDT learns a close approximation to the actual state rewards and the learned rules in DDT are visually interpretable.

Table 2 shows that RL performance of IL reward DDT far exceeds the performance of CRL DDT both when optimal policy is trained using soft reward outputs and when optimal policy is trained using the output of the maximum probability path in the tree. RL performance of IL reward DDT using soft reward is comparable to performance of a deep neural network reward function. In Appendix D, we compare reward DDT in Fig 5 learned from pairwise preferences with a DDT trained with explicit reward labels and a classification loss and find no significant degradation in interpretability from using pairwise preferences. This is encouraging since requiring someone to hand label each individual state with reward values is much more cumbersome than requiring binary preference labels over trajectories.

### 4.2.3 MNIST (0-9) GRIDWORLD

**Setup** To assess scalability of our framework we use a 10x10 gridworld with state space comprising of MNIST digits 0 to 9. We train 2 reward DDTs of depth 4 with simple internal nodes and leaf nodes of type CRL with $\mathbf{R} = (0, 1, .., 9)$, and IL with $R_{\min} = 0$ and $R_{\max} = 9$ respectively.

**Results** Row 3 of Table 2 shows the IL soft reward performance is comparable to a black-box ConvNet learned reward. However, we find that performance of CRL softmax and argmax is significantly degraded, but much better than a random policy. This provides evidence that our framework maintains high performance for much longer horizon and more difficult tasks when using

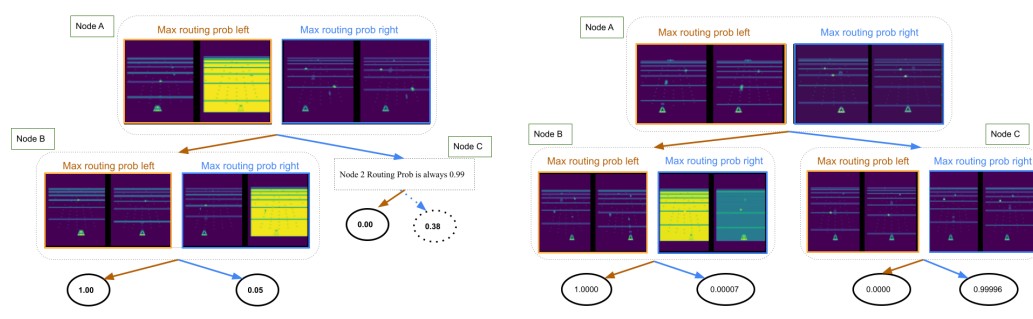

(a) DDT without activation penalty regularization       (b) DDT with activation penalty regularization

Figure 6: **Visualization of Beam Rider Reward DDTs**. We plot the DDTs trained without (a) vs with (b) a regularization penalty on the internal node routing probabilities. We find that the regularization helps the DDT use all leaf nodes, but hurts performance during RL (see Table 3)

interpolated leaf nodes (IL). We analyzed learned DDT and find it still retains explainability despite increase in depth, while still performing comparably to a black-box reward learning approach.

## 4.3 ATARI

To further test the efficacy of learning interpretable rewards, we trained reward DDTs on the Beam Rider and Breakout Atari games (2). Learning rewards for these games is quite challenging as the states are high-dimensional pixel inputs consisting of stacks of 4 $84 \times 84$ video frames.

**Setup** We generate preference training data by generating pairs of trajectory snippets obtained from a variety of partially trained Proximal Policy Optimization (PPO) (54) policies. We follow same procedure proposed by Brown et al. (12) who learned reward functions using deep convolutional neural networks for these games. We then test whether a reward DDT can match the RL performance of the T-REX deep neural network trained by Brown et al. (12), while being interpretable. Because of the complexity of the task, we use sophisticated internal nodes and IL leaf nodes with $R_{\min} = 0$ and $R_{\max} = 1$ (see Appendix E for full details).

To enable interpretation of learned reward DDT, we create a synthetic trace at each internal node. The synthetic trace creates a sequence of states with probability of routing left monotonically decreasing. At an internal node, the trace begins with the state that has maximum probability of being routed left and ends with the state that has minimum probability of being routed left. For ease of visualization, we show the first and last state in the trace.

While trying to create synthetic trace for internal nodes that are children of root node, we discovered that root node's children nodes were using leaf nodes unequally, in the sense that one of child nodes of Node C was never used, subsequently thesub-tree that began at the unused node did not ever see any input state being routed to it. We re-trained the sophisticated reward DDT with the same hyperparameters, but with an added penalty regularization (see Appendix A) and after training, we re-created the synthetic traces for each internal node. We similarly trained a reward DDT with and without penalty for Breakout (see Appendix E). We optimized a policy for each of these learned reward functions by training a PPO agent for 50 million frames and compare it against the previous benchmark in learning from sub-optimal demonstrations, T-REX (Trajectory-ranked Reward EXtrapolation) (12). For training the PPO agent, we utilize each learnt reward DDT in two ways: we either obtain a soft reward over all leaves from tree or we choose the path with maximum routing probability and the reward in this case is obtained by argmaxing over the maximum probability path.

**Results** We visualize synthetic trace for each internal node in the sophisticated reward DDT trained without penalty Fig 6a and compare it against that of reward DDT trained using penalty Fig 6b. In (a) we see that Node A routes states where agent hits an enemy ship to left and states where it misses enemy ships to right. Then Node B routes states where it looks like it will hit an enemy ship to a reward of 1.0 but interestingly routes states where it has hit an enemy ship to a reward of 0 (yellow flash indicates an enemy being destroyed). This allows us to see a misalignment in the learned reward

| | DDT | | | | Baseline T-REX |
| Game | ¬penalty ¬argmax | ¬penalty argmax | penalty ¬argmax | penalty argmax | |
| --- | --- | --- | --- | --- | --- |
| Beam Rider | **5212.04** | 725.84 | 793.12 | 3228.66 | 4742.68 |
| Breakout | **65.48** | 45.21 | 21.28 | 24.16 | 55.18 |

Table 3: **Reinforcement learning using reward DDTs.** We report performance averaged over 100 rollouts. We find that not using a regularization penalty (¬penalty) and allowing a soft output (¬argmax) achieves the best results, even performing slightly better than a large end-to-end neural network reward function (T-REX(12)), trained on the same data.

function. We investigated this further and found that when the agent loses a life, this also triggers a flashing yellow screen. Thus,agent appears to be misinterpreting yellow flash and associating it with a penalty, when it should be associated with a reward. In Fig 6b we see a similar trend for Node B.

We visualize synthetic trace for Breakout unregularized and regularized Reward DDT in Fig 14 and Fig 15. And for both traces, we find at Node A, states that have more number of bricks missing, have a higher routing probability >0.5 and are routed left to Node B while the states in Node A that have no or very few bricks missing have a lower routing probability( routing probability <0.5) and thus are routed right, i.e. to Node C.For more details,see Appendix E In Table 3 we summarize learned policy performance under 4 different scenarios: without Penalty without Argmax, without Penalty with Argmax, with Penalty without Argmax (returning soft reward averaged over all leaf nodes), with Penalty with Argmax for both Beam Rider and Breakout along with T-REX performance on each of these games. We see that RL performance is hurt by adding regularization penalty. We achieve the best scores when using no penalty and a soft output. Interestingly, when using a regularization penalty, using a hard output of the DDT (returning the reward from the leaf node with highest probability) performs best.

## 5 CONCLUSION

We formulated and analyzed a novel method to learn an interpretable reward using differentiable decision trees. Our framework is capable of explaining the most significant features that determine the final reward and routing probability. For low dimensional tasks such as CartPole and MNIST GridWorld environments, our framework is capable of providing global explanations for all inputs. For higher dimensional tasks such as Atari, we approximate global explanations by leveraging aggregations of local explanations by finding the input states that maximally and minimally activate the routing probability of each internal node. On one hand, we provide evidence that reward DDTs are a viable alternative to end-to-end deep network rewards and can perform on-par and sometimes better than their deep neural network reward counterparts; however, for complex domains like Atari, this performance comes at the cost of using DDT in a way that is not interpretable since a soft output that is a weighted sum of outputs of all leaf nodes is hard to interpret. Ideally, we could use reward DDTs with hard reward outputs—the reward output during policy optimization would come from a single leaf node, allowing us to trace the reward output to a small number of binary routing decisions at the internal nodes. While optimizing this kind of hard output (argmax) process works well for the simpler domains we studied (e.g., CartPole and MNIST Gridworlds); it seems to hurt performance on more complex domains. We hypothesize this might be a result of the reward function being too sparse–the number of possible reward outputs is limited, which may adversely affect policy learning. Thus, our results reveal a tension between wanting highly shaped rewards to ensure good RL performance, while also wanting simple, non-shaped rewards to afford interpretability. Future work should investigate this trade-off in more depth.

Our experimental results also provide preliminary evidence that our framework can be used as an alignment debugger tool to inspect learned reward functions for correctness and for capturing the features learned by a model that are misaligned with respect to human intent. We hypothesize that recently proposed methods for human-in-the-loop representation and feature learning (8; 7) and methods for identifying causal features using small amounts of human annotations (25) could enable humans to provide feedback to better align learned reward DDTs.Future work can extend the idea of sophisticated internal node to transformer-based internal node for language and text inputs.

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

## A    DDT ROUTING PENALTY REGULARIZATION

We take inspiration from [19] for adding penalty regularization and we first explain how penalty is defined at each internal node and then elaborate on calculating penalty for a single state over the whole DDT.

The cross-entropy between desired routing probability distribution of an internal node such that it's children nodes are equally used and the actual routing probability distribution is referred to as Penalty and is given by

$$\alpha_i = \frac{\sum_{\mathbf{x}} P^i(\mathbf{x}) p_i(\mathbf{x})}{\sum_{\mathbf{x}} P^i(\mathbf{x})} \tag{3}$$

where the probability of a current internal node is $p_i(\mathbf{x})$ and path probability from root node to an internal node is $P^i(\mathbf{x})$.

Penalty over the whole DDT for a single state is defined as sum over all internal nodes for the given input $\mathbf{x}$

$$C = -\lambda \sum_{i \in \text{ Inner Nodes}} 0.5 \log\left(\alpha_i\right) + 0.5 \log\left(1 - \alpha_i\right) \tag{4}$$

where hyper-parameter $\lambda$ controls the strength of penalty $\lambda$ in reward DDT so that the penalty strength is proportional to $2^{-d}$ and decays exponentially with depth of tree. Finally the penalty term for learning reward tree from pairwise preferences is calculated by taking the mean over all penalties for all states in the pairwise demonstrations.

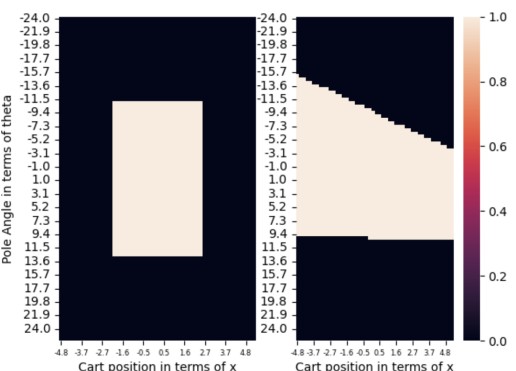

Figure 7: **Cartpole Reward DDT.** Ground Truth Reward on the left compared to Learnt Reward on the right. The reward model learnt by DDT is missing vertical boundaries visually, implying that it failed to pick up on cart position when contrasted with ground truth reward model that has both horizontal and vertical boundaries corresponding to pole angle and cart position respectively.

## B    ADDITIONAL CARTPOLE DETAILS AND ANALYSIS

For learning the reward DDT on classic control CartPole environment Fig 8 , we used a learning rate and weight decay both equal to 0.001 and the Adam optimizer. The neural network we compare to, is trained on same dataset and is comprised of 2 fully connected layers with Leaky ReLU as non-linearity between the layers and the output from the last fully connected layer goes through a sigmoid activation for the final reward from the neural network.

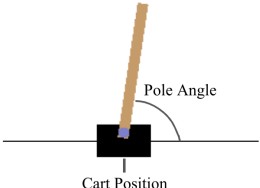

Figure 8: CartPole

To verify that our DDT reward model picks up on the misalignment in the reward function with respect to input features, particular cart position, we also compared the ground truth reward with the

output of the learned reward obtained by taking the argmax class from the leaf node with maximum routing probability. We plot these two rewards as a function of pole angle and cart position in Figure 7

Figure 7 depicts that the learned reward is misaligned: the reward DDT has learned to approximate the preferences over pole angle, but pays much less attention to the pole angle when predicting rewards.

## C  MNIST GRIDWORLD ADDITIONAL DETAILS

In this environment,the action space $a$ contains 4 main actions: go left, go right, move up, move down. The transition function is stochastic and moves the agent in the direction chosen with an $80\%$ probability as long as the action does not take it off of the grid. Actions that would result in leaving the grid result in a self transition.

And the neural network used to learn reward from pairwise human preferences consisted 2 convolutional layers with kernel size 7 and 5 respectively and stride 1 with LeakyRelu as the non-linearities followed by 2 fully connected layers.

**MNIST (0/1) Gridworld Experimental Details**   For training the reward DDT with simple internal nodes and CRL leaf nodes, we use a learning rate of $0.001$, weight decay of $0.05$, and the Adam optimizer (35).

## D  MNIST (0-3) GRIDWORLD ADDITIONAL RESULTS AND ANALYSIS

In this section we provide detailed analysis about interpretability of different DDTs, beginning from comparison between Reward DDT and Classification DDT, then comparing Reward DDTs constructed using two different leaf node formulations, followed by comparison of different regularization on a reward DDT.

Note that for both reward DDTs with different leaf nodes CRL and IL, we trained using a learning rate of $0.001$ and weight decay $0.005$ and the Adam optimizer. And the neural network details are same as defined above in Appendix C.

### D.1  MIN-MAX REWARD INTERPOLATION TREE VS CLASSIFICATION TREE

We train a DDT with explicit reward labels and a classification loss, as in, we re-produce the classification DDT from [19] and compare it to reward tree learned using preferences(refer to Sec 4.2.2 of main paper).

For comparison of reward tree against the classification tree trained using ground truth labels, we plot the heatmaps of internal nodes in both the trees and our results in Figure 9 give evidence that reward tree can capture visual features without any loss in interpretability when compared to the one learnt from simple ground truth labels, even though preferences used here are weaker supervision than ground truth label since preferences used in our experiments are binary as compared to ground truth labels which are 0,1,2,3 corresponding to each actual digit image. This is particularly important in cases where explicit labels are either missing or are hard to be specified or require intensive user-input efforts.

In Figure 9b Node A activates strongly for pixels in the middle of 1s and 2s, routing them left, while and 0s and 3s are routed right. Node B routes left for vertical pixels in the center and sends 2's left and 1's right (note the light shadow looks like a 2 while the darker shadow in the middle that looks like a 1). Node C learns to distinguish between 0s and 3s, routing 3s left and 0s right. This is comparable to the activation heatmaps of the node probability distribution at each of the internal node described for reward tree(in Sec 4.2.2 of main paper).

### D.2  MIN-MAX REWARD INTERPOLATION LEAF DDT VS MULTI-CLASS REWARD LEAF DDT

We train and compare two reward DDTs with simple internal node architecture but with different leaf formulations using the same Bradley-Terry loss over preference demonstration in Figure 10 by

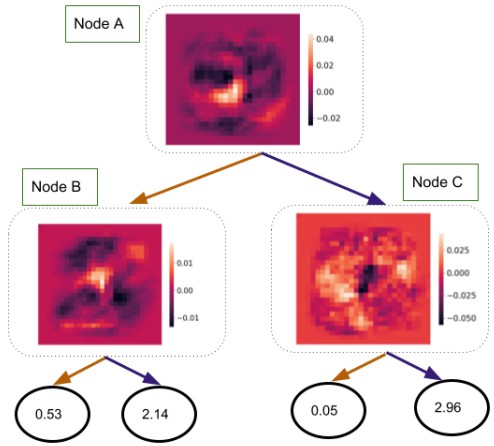

(a) Reward Tree trained using preferences

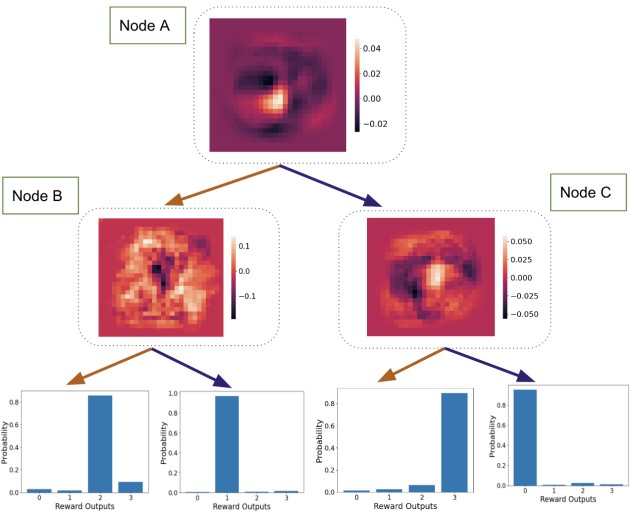

(b) Classification Tree trained on ground truth label

Figure 9: **Visualization of MNSIT (0-3) Reward vs Classification Tree**

visualizing the activation heatmaps of routing probability distributions for the internal nodes and the leaf distribution for each leaf node.

In Figure 10b, each internal node learns to capture almost the same visual feature while the leaf nodes fail to specialize as the argmax output from first two leaf nodes is always a 0 and last two leaf nodes always return a 3. Multi-class Leaf DDT fails to pick up on individual digit in the trajectory , despite requiring the user to input discrete reward vector whereas in the Min-Max Interpolation Leaf DDT each internal node captures different visual attributes and each of the leaf nodes in the interpolated reward DDT is specialized, even though no discrete reward values were given as an input.

This shows that Min-Max Reward Interpolation Leaf DDT is beneficial over Multi-Class Reward Leaf DDT with respect to interpretability and also in terms of human-input efforts. for all states in the pairwise demonstrations.

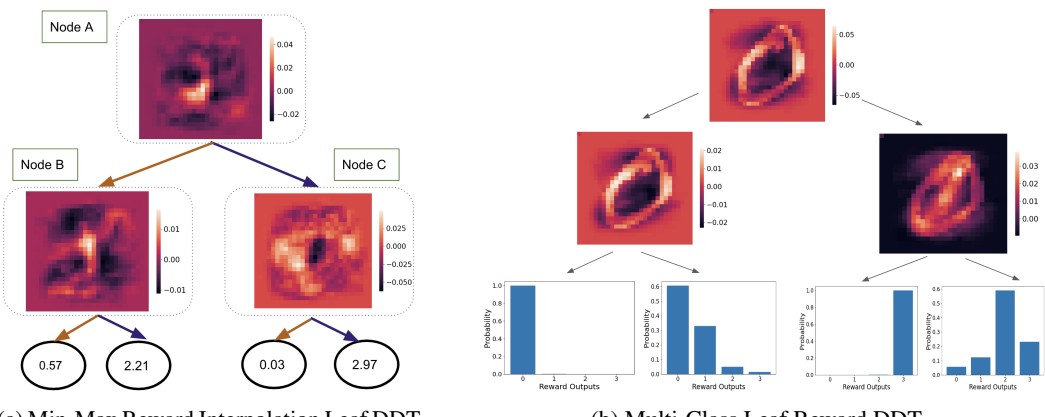

(a) Min-Max Reward Interpolation Leaf DDT

(b) Multi-Class Leaf Reward DDT

Figure 10: **Visualization of MNSIT (0-3) Reward Trees: Min-Max Reward Interpolation Leaf vs Multi-Class Leaf**

### D.3 MIN-MAX REWARD INTERPOLATION DDTS WITH SIMPLE INTERNAL NODES VS SOPHISTICATED INTERNAL NODE

We compare our 2 methods of constructing internal nodes for a reward DDTs.

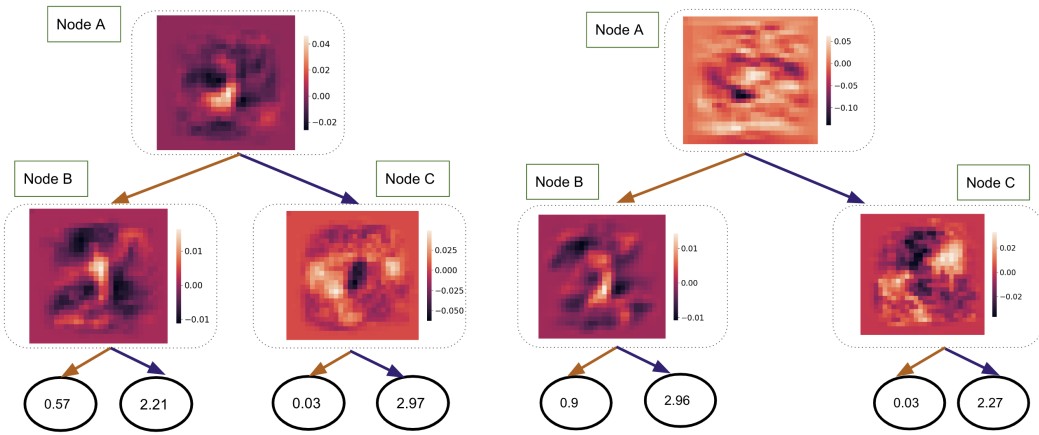

(a) Min-Max Reward Interpolation Leaf DDT with Simple Internal Nodes

(b) Min-Max Reward Interpolation Leaf DDT with Sophisticated Internal Nodes

Figure 11: **Visualization of MNSIT (0-3) Reward Trees :Simple Internal Node vs Sophisticated Internal Node**

Since Min-Max Reward Interpolation Leaf DDT outperforms Multi-Class Reward Leaf DDT, hence we train two different Min-Max Reward Interpolation Leaf DDTs, first one with simple internal nodes and second one with sophisticated internal nodes where a sophisticated internal node contains a single convolutional layer with filter of size 3x3 and stride 1 with Leaky ReLU as the non-linearity followed by the fully connected layer.

In Figure 11b Node A activates strongly for pixels in the middle of 1s and 3s, routing them left, while and 0s and 3s are routed right. Node B routes left for vertical pixels in the center and sends 1's left and 3's right (note the darker shadow in the middle that looks like a 3). Node C learns to distinguish between 0s and 2s, routing 0s left and 2s right. This is comparable to the activation heatmaps of the node probability distribution at each of the internal node described for reward tree(in Sec 4.2.2 of main paper).

Our results depict that in a medium-complexity environment with visual inputs , both DDTs yield relatively equal interpretability but with a higher-complexity environment with larger visual input size such as Atari, the reward DDT with sophisticated node should be used as convolution layer with non-linearity are more powerful in terms of processing an input than a simple fully connected layer.

### D.4 MULTI-CLASS REWARD LEAF DDT REGULARIZATION

Since the DDT with Multi-Class Reward Leaves failed to specialize, this lead us to add the penalty term to the Bradley-Terry preference loss for training the Multi-Class Reward Leaf DDT.

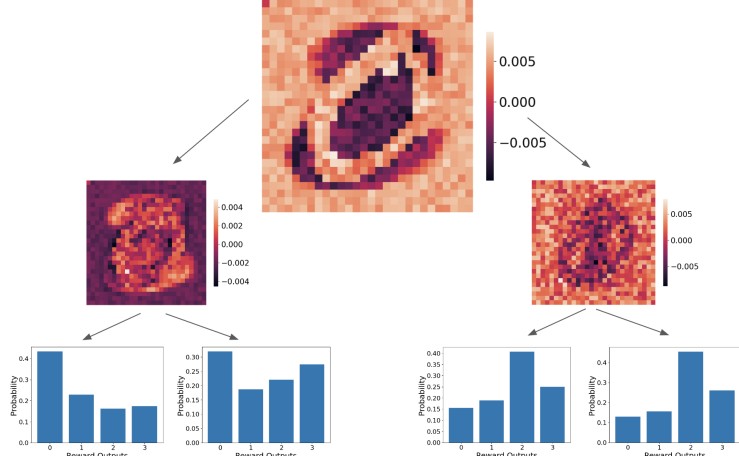

(a) Multi-Class Leaf Reward DDT with penalty calculated over a batch of 50 pairwise preference demonstrations where each demonstration has a single state

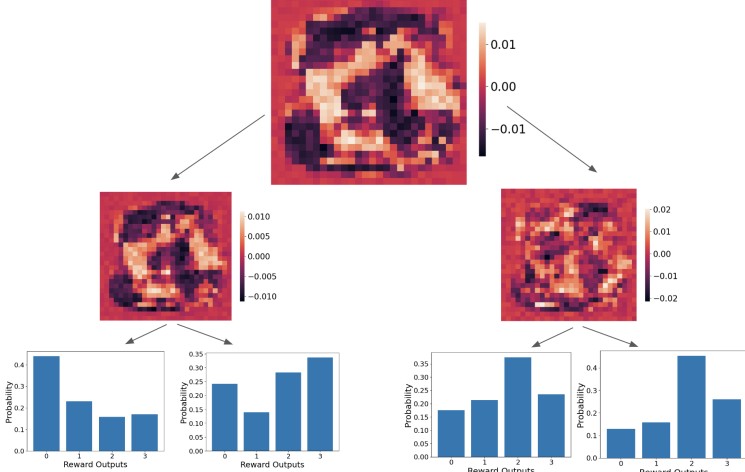

(b) Multi-Class Leaf Reward DDT with penalty calculated over a batch of 50 pairwise preference demonstrations where each demonstration has a single state

Figure 12: Multi-Class Leaf Reward DDT with penalty calculated over different temporal window lengths

For training the Reward DDT,we calculate penalty over batch of 50 pairwise demonstrations where each demonstration contains a single 28x28 greyscale image.To check interpretability, we plot the activation heatmaps of routing probability distributions for the internal nodes and the leaf distribution for each leaf node in Figure 12a and the resulting plots are hugely pixelated, causing a loss in interpretability.

Following this, we increase the temporal window size for calculating penalty, as suggested in [19], and thus we calculate penalty over a pair of 50 preference demonstration where each demonstration is now 50 states long, as opposed to previous case where each demonstration contained a single state. And we again visualize the heatmaps at internal nodes and leaf distributions for each leaf node in Figure 12b. The heatmaps here are little better in contrast to Figure 12a but still have a huge loss of interpretability as compared to Figure 10b.

### D.5 SYNTHETIC TRACE FOR MNIST 0-3 REWARD DDT

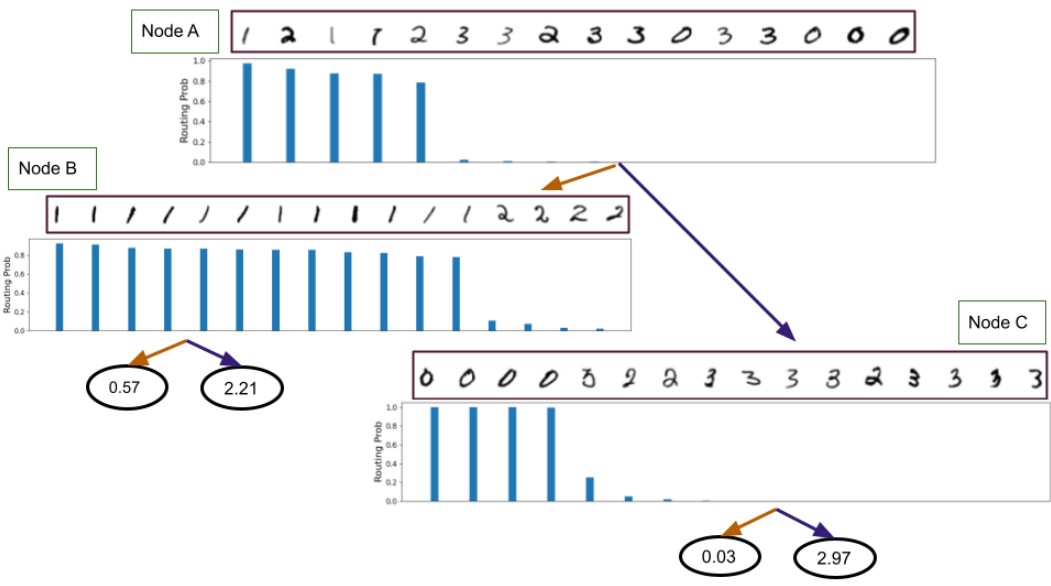

Figure 13: **Synthetic trace of MNIST(0-3) Reward DDT** We plot the trace over the entire input space at each internal node in order of decreasing routing probability from left to right in order to generate global explanations. At each internal node we depict the actual states and their respective routing probabilities. In node B images of 1 are routed to left leaf node as they have higher probability while images of 2 are routed to right leaf node. For Node C, we find that 0s get routed to left leaf node while some 2's which are visually closer to 0s get routed to left leaf node while those 2's that are closer to 3s and actual 3s get routed to right leaf node.For Node A , we find 1s and higher number of 2s are routed to Node B while 0s,3s, and some 2s , which are in terms of pixels, closer in shape to 0s and 3s are routed to Node C.

For the reward DDT trained using Min-Max Interpolation Leaf nodes in Sec 4.2.2 we visualize the synthetic traces fig 13 over the entire state space to generate global explanations. At each internal node, we also plot the respective routing probabilities of every state in the trace.To generate a trace for an internal DDT node, we sort the training data in descending order based on the routing probability assigned to each state. We then sample every Tth state to produce a trace of states that range from those routed most highly to the left child to those most highly routed to the right child.

We find that B images of 1 are routed to left leaf node as their routing probability>0.5 while images of 2 are routed to right leaf node. For Node C, our trace interestingly captures the fact that 0s get routed to left leaf node while some 2's which are visually closer to 0s get routed to left leaf node while those 2's that are closer to 3s and actual 3s get routed to right leaf node.For Node A , we find 1s and higher number of 2s are routed to Node B while 0s,3s, and some 2s , which are in terms of pixels, closer in shape to 0s and 3s are routed to Node C.

Our synthetic trace on basis can explain the reward any state in the entire state space would get in a discrete number of steps.

# E   ATARI

The input to DDT here is a 5-dimensional tensor of size $B \times 2 \times S \times 84 \times 84 \times 4$ where $B$ represents batch size of pairwise preference demonstrations while 2 is represents of number of demonstrations in a pairwise preference and $S$ represents number of states in a single trajectory. The sophisticated internal node architecture here consists of a single convolution layer with kernel of size $7 \times 7$ with a stride of 2 and LeakyRelu as the non-linearity followed by the fully connected linear layer for producing the routing probability inside a tree.We used IL leaf nodes with $R_{\min} = 0$ and $R_{\max} = 1$. Note that we choose these min and max values for simplicity; though the actual numerical value of $R_{\min}$ and $R_{\max}$ can be chosen at the discretion of the user since policies are invariant to positive scaling and affine.

The baseline T-REX, that we compare to has an architecture similar to (15) and consists of 4 convolutional layers of sizes 7x7, 5x5, 3x3 and 3x3 with strides 3,2,1 and 1 respectively, where each convolutional layer has 16 filters and LeakyReLU as non-linearity, followed by a fully connected layer with 64 hidden units and a single scalar output.

**Beam Rider**   For beam rider reward ddt using a batch of $B = 10$ pairwise demonstrations where each demonstration is 25 states long and Adam optimizer, learning rate of $0.0009$ with different seeds. For training Min-Max Reward Interpolation Leaf DDT with sophisticated internal nodes on Beam Rider we tried all combinations possible using the following hyper-parameter settings:

- Seed : 0, 1 and 2
- Batch size of Pairwise Preferences $B$ : 1,10
- Learning Rate: 0.00005, 0.0009 for $B$ equal to 1 and 10 respectively

But for all DDTs trained without the penalty, we ran into the problem of un-equal use of sub-trees.For running-RL and visualizing the synthetic traces we use the reward DDT, trained using seed 0 with B=10 and lr =0.0009.

Note: we use the same exact setting for training the reward DDT with penalty.

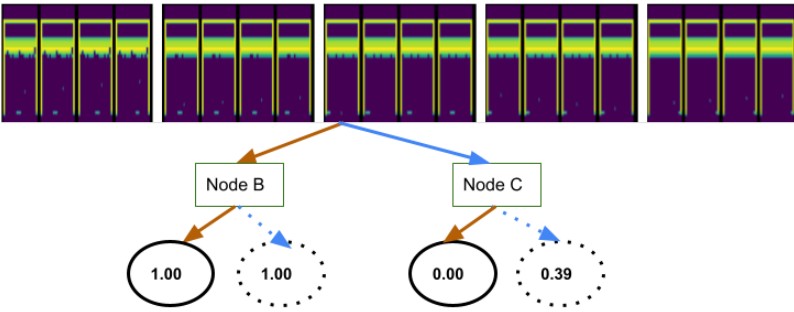

Figure 14: Interpolated DDT of depth 2 trained on Breakout without regularization

**Breakout**   We trained 2 different Interpolated Leaf DDTs on Breakout, one without and another with Penalty added and created the synsthetic trace over all input states starting from states that are routed 100% to left and ending at states that are routed complete right ( as in have 0% probability of beiung routed left).

For Breakout , we show a more rigorous trace than BeamRider,by visualizing states that are routed with 100% , 75%, 50% , 25% and 0% probability to left. And we found that, fig 14 without any penalty added both children nodes of the root node only use their respective left leaves and do not route any state to their respective right leaves. This meant that a synthetic trace could not be visualized for either Node B or Node C.

Next, we visualize the synthetic trace over the penalized DDT, fig 15 and surprisingly , we found that the regularization term added to the final loss of the tree while training was penalizing the DDT so heavily that now the routing probability at Node B and Node C was always between 0.5 and 0.499, which again lead to a failure in being able to create the synthetic trace over inputs as no states were now being routed with a 100% probability neither left nor right and also numerically the difference in routing probability was very trivial.

In fig 14 which depicts our trace for unregularized Reward DDT, we find at Node A, states that have more number of bricks missing have a higher probability(>0.5) of being routed to left, i.e. to Node B which only uses a single leaf node to give a reward of +1 while the states in Node A that have no or very few bricks missing have a lower routing probability( routing probability <0.5) and thus are routed right , i.e. to Node C which again uses a single leaf node to give a reward of 0. We can similarly interpret the trace of regularized reward DDT fig 15 for breakout.

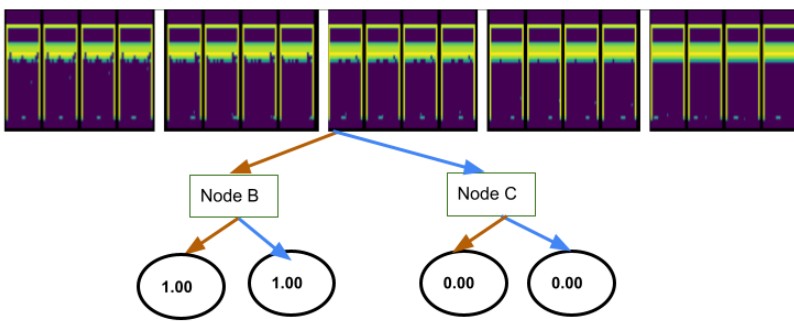

Figure 15: Interpolated DDT of depth 2 trained on Breakout with regularization

