# OpenReview forum: "Can Differentiable Decision Trees Learn Interpretable Reward Functions?"
_ICLR.cc/2024/Conference — Submitted to ICLR 2024_

### Official Review · Reviewer_sBA3 · 2023-10-31

**Soundness:** 2 fair
**Presentation:** 3 good
**Contribution:** 3 good
**Rating:** 3
**Confidence:** 4

**Summary:**

This paper proposes to learn more interpretable reward functions by employing differentiable decision trees. The differentiable decision tree allows for explanations of the reward making decision process, therefore enhancing interpretability. Experiments are conducted on cartpole, a new gridworld MNIST environment, and atari.

**Strengths:**

Strength 1: The idea of improving reward model interpretability is interesting, novel, and timely.

Strength 2: The novel MNIST environment can be of use to the RL community.

Strength 3: The motivation and writing of this paper is quite clear.

**Weaknesses:**

Weakness 1 (Major): One of my main concerns is the lack of repetition over random seeds. In my understanding, the cartpole environment uses 3 seeds (which is not enough for RL), but the Atari environments seem to use only one seed (and treat the seed as a tunable hyperparameter?). This is a cause for concern regarding the replicability and generalization of the proposed method. In addition, it means it is basically impossible to tell if the results about using a regularizer or soft output are significant. The mean and standard deviation across random seeds for each experiment should be reported.

Weakness 2 (Major): The inability to interpret the decision for breakout is a bit worrying. I think this is an important result that should be discussed in the main paper for clarity.

Weakness 3 (Minor): The synthetic trace method could be explained better, as it seems to be a key part of allowing interpretability in complex environments. But it is only introduced in the experiments section. This makes the experiment section a bit hard to follow. I think that introducing this in the methodology section would make more sense.

**Questions:**

Question 1: How would the synthetic trace method be applied to the simple environments (cartpole and mnist)?

Question 2: Could this method be applied to text data?

Question 3: Why does the neural network baseline perform poorly for Cartpole? This is surprising since cartpole is a pretty easy task.

---

> ### Author Response · Authors · 2023-11-21
>
> Thank you for your encouraging feedback and great questions.
> > Random seeds and mean and standard deviation across random seeds for each experiment should be reported.
>
> We have increased the number of seeds for cartpole to 10 and will  update soon the results in the paper with means and standard deviations. We agree that more seeds for Atari is important. You are correct that we only showed results for a single seed (seed 0). We are working on generating results for more seeds for Table 3 and will add these to the camera ready draft.
>
> > The inability to interpret the decision for breakout is a bit worrying. I think this is an important result that should be discussed in the main paper for clarity.
>
> Due to space constraints we put the Breakout results in the appendix, but will work to add more discussion to the main paper. We would like to highlight the fact that the Breakout results are interpretable. In Fig 14,which depicts our trace for unregularized Breakout Reward DDT,  we find at Node A, states that have more number of bricks missing have a higher probability(>0.5) of being routed to left, i.e. to Node B which only uses a single leaf node to give a reward of +1 while the states in Node A that have no or very few bricks missing have a lower routing probability( routing probability <0.5)  and thus are routed right , i.e. to Node C which again uses a single leaf node to give a reward of 0. We can similarly interpret the regularized reward DDT for Breakout.
>
> > The synthetic trace method could be explained better, as it seems to be a key part of allowing interpretability in complex environments.
>
> We agree that the synthetic trace is an important contribution to our work. We will introduce it earlier as suggested.
>
> > How would the synthetic trace method be applied to the simple environments (cartpole and mnist)?
>
> We have generated synthetic traces for MNIST and include them in Appendix D.5.  To generate a trace for an internal DDT node, we sort the training data in descending order based on the routing probability assigned to each state. We then sample every Tth state to produce a trace of states that range from those routed most highly to the left child to those most highly routed to the right child. Interestingly, we find that these traces can uncover interesting edge cases and anomalies. e find that B images of 1 are routed to left leaf node as their routing probability>0.5 while images of 2 are routed to right leaf node. For Node C, our trace interestingly captures the fact that 0s get routed to left leaf node while some 2's which are visually closer to 0s  get routed to left leaf node while those 2's that are closer to 3s and  actual 3s get routed to right leaf node. For Node A, we find 1s and higher number of  2s are routed to Node B while 0s,3s, and some 2s , which are in terms of pixels, closer in shape to 0s and 3s are routed to Node C.
>
> We will also add cartpole traces to the appendix of the camera ready draft.
>
> > Could this be applied to text data?
>
> This is a great question. While out of scope for this paper, we think there may be a possible path forward by further extending the idea of a sophisticated internal node in a reward DDT to a transformer-based or other type of internal node better suited for text inputs. Given a prompt and output, we could feed this into a small transformer node to get it to predict routing probabilities of left and right. It may be possible to use self-attention-based interpretability or local explanations based on counterfactuals [1], then we could try and gain understanding of what parts of the sentence are leading to specific reward outputs.
>
> Another idea for future work would be to take a pretrained LLM that has already been trained via RLHF and then try to distill it into a DDT. The internal nodes could still be transformer nodes or another form of node that can process text such as 1-d convolutions on word embeddings. Thus, the main challenges are how to design an internal node architecture for processing text. These nodes can then be organized hierarchically as a DDT to perform reward learning. We think this is a very exciting area of future work!
>
> [1] Ross et al. "Explaining NLP Models via Minimal Contrastive Editing (MiCE)." ACL-IJCNLP 2021.
>
> > Why does the neural network baseline perform poorly for Cartpole?
>
> We were also confused about these results. Before submission, we tried a wide variety of architectures, but were unable to improve performance. We have looked into this more. Upon further inspection we found the problem. We were basing our implementation on prior work [2] where the reward function output was fed through a sigmoid. However, this seems to significantly impact performance. With no sigmoid, results are much better. We will update Table 1 soon with these new results.
>
> [2] Brown et al. "Extrapolating beyond suboptimal demonstrations via inverse reinforcement learning from observations." ICML, 2019.

---

### Official Review · Reviewer_V4Dp · 2023-10-31

**Soundness:** 3 good
**Presentation:** 3 good
**Contribution:** 3 good
**Rating:** 6
**Confidence:** 4

**Summary:**

This paper delves into the significant issue of learning interpretable reward functions based on human preferences. The authors propose the use of differentiable decision trees (DDT) to model these reward functions and follow the standard reinforcement learning from human feedback (RLHF) pipeline. The paper conducts intriguing experiments across various domains to demonstrate the practicality of these interpretable reward functions in diagnosing misaligned objectives within them.

**Strengths:**

Integrating structural and interpretability constraints into the RLHF pipeline is of paramount importance due to its diagnostic capabilities for misalignment issues. Although the paper doesn't introduce novel algorithms (it employs standard RLHF with DDT reward functions), its empirical investigations are interesting. The experiments, especially in Cartpole, effectively showcase the aforementioned diagnostic utility.

**Weaknesses:**

Limited algorithmic novelty.

**Questions:**

In the related work section, it would be beneficial to discuss research on expert-driven reward design techniques that incorporate structural and interpretability constraints [1, 2, 3].

Regarding Table 1, could the performance of the baseline neural network reward be enhanced by using a richer representation such as a high-capacity deep neural network?

References:

[1] Jiang, et al. Temporal-Logic-Based Reward Shaping for Continuing Reinforcement Learning Tasks. 2021.

[2] Devidze, et al. Explicable Reward Design for Reinforcement Learning Agents. 2021.

[3] Icarte, et al. Reward Machines: Exploiting Reward Function Structure in Reinforcement Learning. 2022.

---

> ### Author Response · Authors · 2023-11-19
>
> Thank you for your encouraging feedback and great questions.
>
> > Limited algorithmic novelty
>
> We would like to clarify the following contributions:
> - Derivation of fully differentiable DT training using pairwise preferences for reward learning. We are the first to demonstrate a method for learning rewards using decision trees in a fully differentiable manner. The idea of soft-routing is common across any kind of DDT, but training a reward function uses a different loss function than prior work on classification. Thus our loss function is one of our paper's contributions. We cannot simply use a standard classification loss because we do not assume access to reward labels for every state.
> - Characterization and evaluation of different leaf nodes. As discussed in Section 3.2 of our paper, prior work on DDTs has mainly focused on classification tasks and we borrow the multi-class reward leaf formulation from prior work. Our results show that using classification style leaf nodes do not work as well as our proposed min-max interpolation leaf nodes which we find is best suited for reward DDT learning.
> - Explanations per leaf node: pixel based attribution as well as synthetic traces. Our focus on explainable rewards via tree structures and on using both synthetic traces and pixel-based attribution to enable reward function debugging.
> - First approach to scale tree-based reward learning to visual inputs.
> - Characterization and evaluation of RL-time reward predictions: soft vs hard. Interestingly, we find there is a tension between wanting hard routing for interpretability and wanting soft routing to enable better shaped rewards for reinforcement learning.
>
> We would like to emphasize that while we use and extend some of the prior tools developed for DDT training for classification tasks, our experimental results are highly novel as we are the first to study training DDTs for reward learning to enable better explainability and alignment verification. While learning rewards from preferences is common (e.g. RLHF), prior work is not interpretable. Thus, we believe our contributions will be of great interest to the community.
>
> > Related work incorporating structural and interpretability constraints [1, 2, 3].
> [1] Jiang, et al. Temporal-Logic-Based Reward Shaping for Continuing Reinforcement Learning Tasks. 2021.
> [2] Devidze, et al. Explicable Reward Design for Reinforcement Learning Agents. 2021.
> [3] Icarte, et al. Reward Machines: Exploiting Reward Function Structure in Reinforcement Learning. 2022.
>
> Thank you for pointing out this related work. We will add these papers to our related work. The referenced papers are similar to our paper in that they also have a focus on structured and interpretable rewards. However, there are also some key differences.
>
> Jiang et al. [1] develop a novel theory for reward shaping for average reward RL tasks and incorporate temporal logic specifications to help with RL learning speed. The major distinction between this work and ours is that Jian et al. assume access to true reward and seek to add a shaping reward, whereas we do not assume any access to the ground truth reward, nor to a temporal logic shaping reward signal. Instead we must learn the entire reward function purely from pairwise preferences. One exciting area of future research could be to investigate how to use a user-supplied LTL formula to help guide building the DDT. Another area would be to try and distill a DDT into an LTL specification.
>
> Devidze et al. [2] focus on finding rewards that are sparse and yet speed up policy learning. This problem is analyzed from an algorithmic teaching perspective where a teacher is assumed to have access to a ground-truth reward function and optimal policy for that reward function and must distil the reward function into something that is informative (leads to fast RL) but also sparse. By contrast, our approach focuses on learning rewards that are initially unknown and unspecified. Given a known reward that is not sparse, it may be possible to distill the reward into a DDT to obtain a sparser, but still informative reward. We think this is an interesting area of future work.
>
> Icarte et al. [3] focus on helping with reward design, not reward learning from human feedback. They assume that an RL agent is given access to the specification of the reward function in the form of a finite state machine reward. Icarte et al. then contribute is a collection of RL methods that can exploit a reward machine’s internal structure to improve sample efficiency and learn policies in a more
> sample efficient manner. This paper presents a nice formalism for thinking about how to design interpretable rewards that are well suited for RL, but, similar to the other papers, assumes the human can pre-specify this reward. We focus on the problem of using sparse human feedback to teach an AI the reward function rather than needing to specify it manually.

---

> > ### Author Response · Authors · 2023-11-19
> >
> > > Table 1 neural network results
> >
> > We were also confused about these results. Before submission, we tried a wide variety of architectures, but were unable to improve performance. We have looked into this more. Upon further inspection we found the problem. We were basing our implementation on prior work [4] where the reward function output was fed through a sigmoid. However, this seems to significantly impact performance. With no sigmoid, results are much better. We will update Table 1 soon with these new results.
> >
> > [4] Brown et al. "Extrapolating beyond suboptimal demonstrations via inverse reinforcement learning from observations." ICML, 2019.

---

### Official Review · Reviewer_wNaa · 2023-11-01

**Soundness:** 2 fair
**Presentation:** 3 good
**Contribution:** 2 fair
**Rating:** 5
**Confidence:** 3

**Summary:**

This paper proposes to represent the reward function in RL as a decision tree where the nodes and leaves are expressed as neural network layers. At each node, a binary classification is conducted to select which child node to visit. At the leave nodes, a multi-classification module and a regression module are adopted to generate discrete and continuous reward value respectively. This tree structure can provide interpretability in determining which behavior should be rewarded or penalized. The experimental section validates the proposed approach with comparisons with different baselines in multiple benchmarks.

**Strengths:**

* `Originality`: the idea is original.

* `Quality`: there is no major technical issue.

* `Clarity`: the fundamental ideas in this paper are explained.

**Weaknesses:**

The idea of inserting nonlinear layers in the tree appears to contradict the purpose of using the tree structure for interoperability. It would be great if the author explained the fundamental difference between this tree and a tree-like neural network. In other words, is it possible to design a neural network that connects the neurons as in the tree and uses ReLUs for branching? If the Author agrees, the author can clarify what are the benefits of using the tree structure instead of NN. Does it reduce complexity, etc.  The author also does not explain how to determine the structure of those trees. Do those trees grow like XGBoost trees? According to the experimental section, the trees are not deep. The baseline NNs are not deep, either. The motivation for trading performance for interoperability is not very strong.

**Questions:**

Please address my question in the `Weakness` field.

---

> ### Author Response · Authors · 2023-11-19
>
> Thank you so much for your encouraging and valuable feedback.
>
> > Inserting nonlinear layers in the tree appears to contradict the purpose of using the tree structure for interoperability.
>
> We assume the reviewer means “interpretability”. If not, we would appreciate clarification by what is meant by “interoperability” in this context.
>
> Non-linearities are a core part of even classical DTs that are known for interpretability. We investigate two approaches: simple internal node and sophisticated internal node. For the simple internal node we use a sigmoid function to essentially perform logistic regression. While in practice one can always collapse an input into a single vector and use a sigmoid, this loses much of the structure of higher-dimensional inputs in more complex images. Convolutions are known to better represent image features than taking a linear combination of stacked pixels. We only use a single convolutional layer with 7x7 kernel size, stride 2 and padding 0. This maintains better interpretability than a fully convolutional approach while giving more expressive power to the reward DDT for more complex image observations.
>
> With respect to interpretability, we mainly achieve this through the hierarchical nature of the reward tree. By breaking down a complex reward prediction problem into a sequence of simpler decision problems, the goal is to create a digestible and traceable path toward reward predictions. It is the tree structure that we seek to leverage for studying interpretable reward functions. We agree that there is a tension between making nodes more complicated to achieve better performance and enabling interpretability. We will add a discussion of this to our draft.
>
>
>
> > Explain the fundamental difference between this tree and a tree-like neural network...Does it reduce complexity?
>
> Tree-like neural networks, are primarily neural nets with the defined routing mechanisms for the input. In these structures, the neurons at each level of neural network see the input only after it’s been transformed from the previous layer and all learnable parameters of the network are dependent on input. Whereas in our framework, we learn hierarchical filters that are all independent of each other such that each tree level, every internal node sees the whole raw input and not the transformed input and our leaves are not dependent on the input. We fail to understand how routing can be done using ReLU  except in case of binary routing/neural networks with only 2 neurons in each layer, as ReLU is unbounded and this fails the premise of conditional probabilistic routing, which is a key property of decision trees.
> Using trees over neural networks not only lends interpretability as trees but also it ensures conditional computation where each sample is only processed through a small set of nodes based on routing probability as opposed to neural networks where all neurons in every layer process every input. This also implies training and inference speed-ups in trees compared to neural networks [1, 2].
>
> [1] Hazimeh et al.  “The tree ensemble layer: Differentiability meets conditional computation” ICML 2020.
>
> [2] Tanno et al. “Adaptive neural trees.” ICML 2019.
>
> > The author also does not explain how to determine the structure of those trees.
>
> We focused our analysis on shallower trees as there is a trade-off between depth and interpretability, even for classical DTs [4]. In practice, the use of a validation set could be used to determine a good tree depth. Because we are concerned with whether we can learn interpretable DDTs we chose to focus on the shallower trees shown in the paper; however, we did use depth 4 DDTs for MNIST (0-9) and found it worked well. The expressivity of the reward does depend on the tree depth and the expressivity-interpretabiilty trade-off should be taken into consideration by a designer.
>
> > Do those trees grow like XGBoost trees?
>
> No we do not grow the trees. This would make things non-differentiable which would violate one of the goals of our paper. However, it is an interesting area of future work.
>
> > According to the experimental section, the trees are not deep. The baseline NNs are not deep, either.
>
> The baselines are chosen using state-of-the-art deep neural networks from prior works [1-3]. We did not feel the need to make these networks even larger since they were presumably carefully tuned in prior work.
> In general the larger the decision tree the less interpretable it is [4]. Thus, we tend towards smaller trees to stress test the idea of interpretable reward functions (which should be shallow).
>
> [1] Christiano et al. "Deep reinforcement learning from human preferences". In NeurIPS 2017.
>
> [2] Ibarz et al. "Reward learning from human preferences and demonstrations in atari." NeurIPS 2018.
>
> [3] Brown et al. "Extrapolating beyond suboptimal demonstrations via inverse reinforcement learning from observations." ICML 2019.
>
> [4] Molnar, C. (2020). Interpretable machine learning.

---

> > ### Author Response · Authors · 2023-11-21
> >
> > > The motivation for trading performance for interoperability is not very strong.
> >
> > Again, we assume the reviewer means “interpretability”. Please let us know if you do mean “interoperability”.
> >
> > We agree that it is important to motivate interpretability. In many cases [1]  prior work has shown that you do not have to sacrifice performance for interpretability. However, we believe there is often going to be a tradeoff. Most deep learning work focuses on training larger and larger models. We take the opposite approach and seek to evaluate whether smaller and more interpretable models can also do the job. We find that our framework is often capable of explaining the most significant features that determine the final reward.
> >
> > [1] Rudin, Cynthia. "Stop explaining black box machine learning models for high stakes decisions and use interpretable models instead." Nature Machine Intelligence, 2019

---

### Official Review · Reviewer_qpnV · 2023-11-01

**Soundness:** 3 good
**Presentation:** 3 good
**Contribution:** 3 good
**Rating:** 6
**Confidence:** 4

**Summary:**

In reinforcement learning from human feedback, the learned reward model is usually a neural model, which is usually not interpretable. This work proposes learning interpretable reward models represented by Differentiable Decision Trees (DDTs). Empirically, DDTs can learn reward models that are useful for RL.

**Strengths:**

This paper novelly uses DDT to learn and represent a reward model, and shows that the method is effective on CartPole, MNIST, and Atari domains. Using a tree-based reward model is overall an inspiring approach that is under-explored in the literature.

**Weaknesses:**

The paper started by using learning from human feedback as a motivating example (Fig. 1). However, the domains are all simulated domains. Since rewards in simulated domains are human-designed, it is comparatively simple to learn these reward functions.

Clarity on the contributions. It would be helpful to clarify which part of the algorithm is exactly the DDT algorithm itself, and which part is its adaption to reward learning. So it is clear to see the contributions of this work.

**Questions:**

Is it practical to learn interpretable reward functions for more realistic domains, like reinforcement learning from human feedback for large language models?

---

> ### Author Response · Authors · 2023-11-18
>
> Thank you for your encouraging feedback and great questions.
>
> > Since rewards in simulated domains are human-designed, it is comparatively simple to learn these reward functions.
>
> We would like to emphasize that even human-designed rewards are often challenging to learn accurately. Using RL environments for studying imitation and reward learning by making the true reward unobserved is common practice for many papers published at NeurIPS, ICML, and ICLR, eg. [1-5]. Using synthetic preference data and synthetic demonstrations for testing enables easier evaluation and comparison with other approaches. It also allows performance evaluation relative to an oracle (RL policy trained on true reward), which is much more difficult if the reward comes implicitly from a human. As such, we do not think this is a major limitation, but agree that exploring user studies with real human preferences is an exciting area for future work.
>
> [1] Fu et al. "Learning Robust Rewards with Adversarial Inverse Reinforcement Learning." ICLR 2018.
>
> [2] Brown et al. "Safe imitation learning via fast Bayesian reward inference from preferences." ICML 2020.
>
> [3] Lee et al.  "PEBBLE: Feedback-Efficient Interactive Reinforcement Learning via Relabeling Experience and Unsupervised Pre-training." ICML 2021.
>
> [4] Lee et al. "B-Pref: Benchmarking Preference-Based Reinforcement Learning." NeurIPS 2021.
>
> [5] Tien et al. "Causal Confusion and Reward Misidentification in Preference-Based Reward Learning." ICLR 2022.
>
> > Clarity on contributions
>
> We would like to clarify the following contributions:
> - Derivation of differentiable DT training using pairwise preferences for reward learning. We are the first to demonstrate a method for learning rewards using decision trees in a fully differentiable manner. The idea of soft-routing is common across reward DDTs and prior work on classification DDTs but training a reward function uses a different loss function which is one of our paper's contributions. We cannot simply use a standard classification loss because we do not assume access to reward labels for every state.
> - Characterization and evaluation of different leaf nodes. As discussed in Section 3.2 of our paper, prior work on DDTs has mainly focused on classification tasks and we borrow the multi-class reward leaf formulation from prior work. Our results show that using classification style leaf nodes do not work as well as our proposed min-max interpolation leaf nodes which we find is best suited for reward DDT learning.
> - Explanations per leaf node: pixel based attribution as well as synthetic traces. Our focus on explainable rewards via tree structures and on using both synthetic traces and pixel-based attribution to enable reward function debugging.
> - First approach to scale tree-based reward learning to visual inputs.
> - Characterization and evaluation of RL-time reward predictions: soft vs hard. Interestingly, we find there is a tension between wanting hard routing for interpretability and wanting soft routing to enable better shaped rewards for reinforcement learning.
>
> We would like to emphasize that while we use and extend some of the prior tools developed for DDT training for classification tasks, our experimental results are highly novel as we are the first to study training DDTs for reward learning to enable better explainability and alignment verification. While learning rewards from preferences is common (e.g. RLHF), prior work is not interpretable. Thus, we believe our contributions will be of great interest to the community.
>
> > Is it practical to learn interpretable reward functions for more realistic domains, like reinforcement learning from human feedback for large language models?
>
> This is a great question. While out of scope for this paper, we think there may be a possible path forward by further extending the idea of a sophisticated internal node in a reward DDT to a transformer-based or other type of internal node better suited for text inputs. Given a prompt and output, we could feed this into a small transformer node to get it to predict routing probabilities of left and right. It may be possible to use self-attention-based interpretability or local explanations based on counterfactuals [6], then we could try and gain understanding of what parts of the sentence are leading to specific reward outputs.
>
> Another idea for future work would be to take a pretrained LLM that has already been trained via RLHF and then try to distill it into a DDT. The internal nodes could still be transformer nodes or another form of node that can process text such as 1-d convolutions on word embeddings. Thus, the main challenges are how to design an internal node architecture for processing text. These nodes can then be organized hierarchically as a DDT to perform reward learning. We think this is a very exciting area of future work!
>
> [6] Ross et al. "Explaining NLP Models via Minimal Contrastive Editing (MiCE)." ACL-IJCNLP 2021.

---

### Author Response · Authors · 2023-11-22

We sincerely thank the reviewers for their extensive, encouragaing feedback and thoughtful questions. We are really happy and excited that the reviewers believe that we take an inspiring, original and novel  approach for learning interpretable tree-based reward models (Reviewers qpnV,wNaa,sBA3) and that the problem we study is very important in RLHF pipeline  and our experiments across various domains demonstrate the practicality of the interpretable reward functions in diagnosing misaligned objectives within them (Reviewer V4Dp).

We are really grateful to the reviewers for their helpful suggestions and believe they have significantly strengthened the paper. We have updated our draft/pdf  by :
- including references to expert-driven reward design techniques that incorporate structural and interpretability constraints, in paragraph titled Explaining and Interpreting Reward Function under Section 2
- reporting Mean and Standard deviation in Table 1 under section 4.1 across 10 seeds compared to Inter-Quartile Mean we had across 3 seeds before, for DDT(both soft and hard/argmax routing) as well as neural network. We fixed the neural network from before.
- incorporating explanation for Breakout Reward DDT in Results in Sec 4.3 in main paper  and in Appendix E
-  generating synthetic traces for MNIST and including them in Appendix D.5.
-  updated our Conclusion to account for possible extensions to language and text inputs in Section 5

Here we only answer the primary suggestions, questions, and concerns raised by the reviewers.  For our detailed answers to each individual review, please refer to other comments addressed for each specific review.

We have not been able to complete all requested experiments on multiple seeds on Atari due to time and compute power constraints, but we are confident we will be able to add more seeds by the camera-ready deadline. We also want to mention that we will release the code for reproducing our framework and experiments by camera-ready deadline.

Please let us know if you have any questions, concerns or doubts about our framework and experiments,  and we deeply apologize that completing the rebuttal took a long time, but that being said, we would love to interact with the reviewers and answer any questions/address any concerns they might have until the deadline!

---

### Meta-Review · Area_Chair_VH2s · 2023-12-05

**Metareview:**

There was a bit of disagreement among reviewers for this work, but none were enthusiastic in favor of this submission, especially compared to other work submitted at this conference.

My own evaluation is aligned with these ratings.

**Justification For Why Not Higher Score:**

As stated above, the reviewers and myself agree that this submission, in its current state, does not meet the bar of acceptance for this conference, in terms of conceptual novelty or impressive experimental results.

**Justification For Why Not Lower Score:**

N/A

---

### Decision · Program_Chairs · 2024-01-16

Reject